# UNIFIED 3D SCENE UNDERSTANDING THROUGH PHYSICAL WORLD MODELING

**Wanhee Lee**[1]*    **Klemen Kotar**[1]*    **Rahul Mysore Venkatesh**[1]*    **Jared Watrous**[1]*
**Honglin Chen**[2]*    **Khai Loong Aw**[1]    **Daniel L. K. Yamins**[1]

[1]Stanford University     [2]OpenAI

## ABSTRACT

Understanding 3D scenes requires flexible combinations of visual reasoning tasks, including depth estimation, novel view synthesis, and object manipulation, all of which are essential for perception and interaction. Existing approaches have typically addressed these tasks in isolation, preventing them from sharing a common representation or transferring knowledge across tasks. A conceptually simpler but practically non-trivial alternative is to unify these diverse tasks into a single model, reducing different tasks from separate training objectives to merely different prompts and allowing for joint training across all datasets. In this work, we present a physical world model for unified 3D understanding and interaction (**3WM**), formulated as a probabilistic graphical model in which nodes represent multimodal scene elements such as RGB, optical flow, and camera pose. Diverse tasks emerge from different inference pathways through the graph: novel view synthesis from RGB and dense flow prompts, object manipulation from RGB and sparse flow prompts, and depth estimation from RGB and camera conditioning, all zero-shot without task-specific training. **3WM** outperforms specialized baselines without the need for finetuning by offering precise controllability, strong geometric consistency, and robustness in real-world scenarios, achieving state-of-the-art performance on NVS and 3D object manipulation. Beyond predefined tasks, the model supports composable inference pathways, such as moving objects aside while navigating a 3D environment, enabling complex geometric reasoning. This demonstrates that a unified model can serve as a practical alternative to fragmented task-specific systems, taking a step towards a general-purpose visual world model.

## 1 INTRODUCTION

Understanding 3D scenes from visual data is a fundamental challenge in computer vision and an important step toward building world models that support real-world interaction. Holistic 3D understanding requires perceiving visible surfaces and reasoning about hidden geometry in a physically consistent way. For example, to grasp a toy or a cat, one must infer the back or underside that is not directly visible. An agent navigating a cluttered hallway must estimate depth for free space, reason about occluded structure, and manipulate objects to move them out of the way when necessary. These scenarios highlight that 3D perception depends on flexible combinations of reasoning modes, and that a general vision model capturing this broader structure can provide a stronger foundation than systems designed narrowly around specific benchmarks.

Three essential components of 3D understanding are depth estimation, novel view synthesis, and object manipulation, as they collectively capture the challenges of perceiving geometry, inferring unseen views, and reasoning about dynamic interactions. Existing approaches address these tasks independently within specific frameworks and finetuned on narrow datasets, which leaves each model with its own limitations. Depth models cannot infer occluded regions Yang et al. (2024); Wang et al. (2024a), novel view synthesis methods struggle with geometric consistency and precise control Sargent et al. (2024); Yu et al. (2024), object manipulation models Pandey et al. (2024); Wu et al. (2024) allow localized editing but do not extend to scene-level reasoning, and none of them

---

*Equal contribution.

can flexibly handle tasks that they were not optimized for. An alternative is to adopt a unified framework that trains on a superset of all tasks together, free from fixed input–output designs and dataset constraints. In such a framework, supervision from diverse tasks and datasets may shape a representation grounded in the physical scene, enabling knowledge transfer across tasks and supporting robust generalization.

Unifying the diverse components of 3D understanding into a single model that works robustly in real-world scenarios is difficult because it requires both flexibility and controllability. Most previous approaches are restricted to a single form of input and output, for example, mapping RGB images to depth, and therefore cannot flexibly support the diverse queries that arise in 3D understanding. Additionally, achieving strong controllability in generative models has been challenging, motivating extensive research on incorporating additional conditioning signals and architectural modifications to better steer the generation process, most notably exemplified by ControlNet Zhang et al. (2023). Yet reliably enforcing such controls in vision models remains unsolved, particularly for 3D understanding tasks, where outputs often deviate from the intended specification or show distortions in object and scene appearance despite ongoing efforts to address these issues Zhou et al. (2025); Shi et al. (2024). These limitations motivate the need for a new general framework that unifies tasks through flexible prompting while ensuring precise geometric control and physical consistency.

In this work, we propose a physical world model for 3D understanding (3WM) that provides a unified framework supporting diverse inference pathways while remaining precisely controllable. We formulate the model as a probabilistic graphical model (PGM) in which nodes represent multimodal scene elements such as RGB patches, optical flow patches, and camera pose. The graphical model formulation allows traversal across modalities and spatial locations, making it possible to construct flexible inference pathways guided by conditioning. Queries are represented explicitly as tokens rather than being hidden in auxiliary modules, providing a unified physical prompting interface. To make this PGM practical and scalable, we formulate it as an autoregressive next token predictor, enabling efficient training and inference on large-scale data. This framework enables the model to integrate knowledge across tasks and modalities and develop a coherent understanding of the 3D world.

With this design, tasks such as novel view synthesis, object manipulation, and depth estimation emerge naturally within the same system as different forms of zero-shot causal inference rather than as predefined objectives. Through extensive evaluation, we find that our model achieves precise controllability, strong geometric consistency, and robustness in real-world scenarios, outperforming specialized baselines in NVS and 3D object manipulation. Moreover, the model supports flexible geometric reasoning, including joint object manipulation with NVS, complex egocentric navigation, revealing occluded geometry by removing attached objects, and handling depth uncertainty, all of which are required for reliable interaction and navigation in complex real-world 3D environments. These results suggest that 3WM develops a strong 3D understanding of the world and opens the possibility of tackling a wide range of 3D vision problems within a single framework, taking a step toward general purpose visual world models.

Our work makes the following core contributions to unified physical world modeling.

- We introduce 3WM, a unified physical world model that jointly represents RGB, optical flow, and camera pose within a single generative autoregressive framework with a shared prompting interface.

- We propose a local random access sequence formulation that allows the model to condition on, query, and update arbitrary spatial regions, enabling flexible inference pathways in which modalities and spatial locations can be decoded in any order within a GPT-style autoregressive transformer.

- 3WM performs novel view synthesis, 3D object manipulation, and self-supervised depth estimation in a zero-shot manner while achieving state-of-the-art performance on real-world benchmarks.

- The model further supports flexible geometric reasoning capabilities, including compositional camera and object motion, amodal completion, and reasoning about depth uncertainty.

## 2 RELATED WORKS

**Novel View Synthesis** (NVS) has been widely studied as a fundamental task in 3D vision. Regression-based methods Yu et al. (2021); Charatan et al. (2024); Kulhánek et al. (2022); Sajjadi et al. (2022) perform well for view interpolation but yield blurry reconstructions in unobserved regions. This leads to a shift toward generative models, mostly diffusion-based methods, which enable high-quality and diverse NVS. Zero-1-to-3 Liu et al. (2023), trained on large-scale synthetic datasets Deitke et al. (2023); Chang et al. (2015), predicts novel views from a single image using implicit camera conditioning. ZeroNVS Sargent et al. (2024) integrates Zero-1-to-3's approach with a score distillation sampling framework Poole et al. (2022), and extends the application to real-world scenes. Other approaches, such as MotionCtrl Wang et al. (2024c), inject camera embeddings to guide video diffusion without explicit 3D representations. ViewCrafter Yu et al. (2024) utilizes point-cloud rendering using DUSt3R Wang et al. (2024a) for improved performance with better camera motion control. SEVA Zhou et al. (2025) handles diverse novel view synthesis tasks and produces temporally consistent samples and long videos without requiring 3D distillation. In this work, we explore autoregressive sequence modeling for the NVS problem as an alternative to diffusion-based approaches to overcome the limitations of previous works.

**3D Object Manipulation** While NVS focuses on generating novel views of the input scene, object manipulation refers to the task of transforming objects in the scene while keeping the camera fixed. Drag-based image editing methods Wang et al. (2024c); Wu et al. (2024); Shi et al. (2024); Yin et al. (2023) aim to solve this problem by parameterizing object transforms as 2D motion vectors which are then used as conditioning to fine-tune stable diffusion (SD) Rombach et al. (2022). These methods can be naturally extended to more complex 3D transforms by incorporating depth information into the drag vectors Wang et al. (2024b). Another class of models Pandey et al. (2024); Koo et al. (2025), performs 3D object manipulations by editing input depth maps according to the desired object transform and utilizing a depth-conditioned diffusion model to generate the edited image. However, these methods heavily rely on inverting the input image into the SD latent space, which often fails on real-world images Mokady et al. (2023).

**General Motion Control** Recently, several works have shown that motion-conditioned diffusion models can be used to perform sophisticated image manipulations. Geng et al. (2024); Koroglu et al. (2024); Jin et al. (2025) train a spatio-temporal trajectory-conditioned control net on top of a large video diffusion model Bar-Tal et al. (2024). The model demonstrates emergent capabilities such as object and camera control and drag-based image editing and motion transfer. Another set of recent works Gu et al. (2025); Zhang et al. (2024); Feng et al. (2024) uses 3D point trajectories providing more powerful control over image generation. However, they have not shown robust and precise control in real-world scenarios through extensive evaluations. It is also worth noting that our approach is in the same spirit as sequential generative modeling frameworks such as Bai et al. (2024), which can perform various perception and generation tasks using visual prompts. Our approach goes further by enabling random access for scalability, incorporating non-visual queries such as camera pose, and providing precise controllability for 3D tasks.

**Language-Driven Semantic 3D Understanding** Recent multimodal 3D LLMs, such as Inst3D-LMM Yu et al. (2025), Video-3D LLM Zheng et al. (2025), PointLLM Xu et al. (2024), and SceneLLM Fu et al. (2024), focus on semantic reasoning over pre-reconstructed 3D inputs, addressing tasks including 3D dense captioning, visual grounding, and question answering. These models interpret an already constructed 3D scene rather than inferring physical geometry from images or predicting how future observations change under camera or object motion. Their goals, inputs, and evaluation settings therefore differ fundamentally from our formulation, which centers on learning physical 3D structure and transformations directly from 2D observations.

## 3 METHODS

We first introduce 3WM as a probabilistic graphical model $\Psi$ over RGB, optical flow, and camera pose, represented using strictly local HLQ patch codes. We then describe how these variables are serialized into a Local Random Access Sequence (LRAS) of pointer–value tokens, which allows $\Psi$ to learn conditional predictions over arbitrary nodes in the graph. Building on this formulation, we next present our flexible inference pathways, with optical flow acting as a controllable intermediate representation. Finally, we show how these pathways enable zero-shot prompts for 3D tasks, includ-

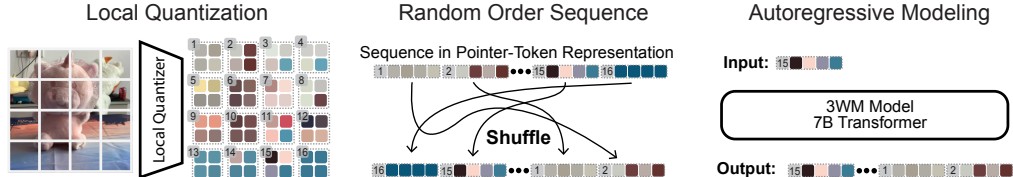

**Figure 1: Local random access sequence modeling.** Our modeling framework has three key components: (a) a local patch quantizer trained based on a small convolutional autoencoder; (b) a video serialization process based on a "pointer-content representation", which allows arbitrary ordering of the patches during training and generation; and (c) an LLM-like autoregressive transformer to predict the contents of the next patch, trained in random sequence order.

ing camera-driven flow prediction for depth from motion, dense flow conditioning for novel view synthesis, and object-specific flow fields for 3D object manipulation.

## 3.1 LEARNING A PROBABILISTIC GRAPHICAL MODEL VIA LOCAL RANDOM ACCESS SEQUENCE MODELING

We introduce a unified, precisely controllable world model that treats visual data as nodes in a Probabilistic Graphical Model (PGM) Koller & Friedman (2009). To efficiently implement such a model, we formulate it as a GPT-style next token predictor through the use of two key innovations: (i) a *local* quantizer that preserves strict patch independence, and (ii) *pointer tokens* that allow random-access encoding/decoding. Together, they let us phrase a wide range of geometric tasks as LLM-style prompts—without task-specific heads, losses, or datasets—while maintaining precise, patch-level control.

**Probabilistic Graphical Modeling.** We construct a learnable PGM ($\Psi$) over local visual variables (e.g., RGB or flow patches) and global control variables (e.g., camera pose). Each variable is assigned a unique pointer address (spatiotemporal patch index) from the set $P$ and contains a value from the discrete codebook $V$. We model the function $\Psi$, which takes as input $X$—the set of variables (pointer-value pairs) observed so far—and an as yet unobserved pointer $p$ from the set $P$, and outputs the distribution over possible values for node $p$ from the codebook $V$. For example, $X$ might contain all patches from the current frame plus a sparse subset from the next frame, while $p$ could be any masked patch in the next frame that $\Psi$ must predict from this partial observation.

$$\Psi : (X,\ p \notin \mathrm{dom}(X)) \mapsto \{\Pr[(p,v)\,|\,X] : v \in V\}.$$

This enables the model to predict the distribution over values at any node and sample values either simultaneously or through autoregressive sampling, effectively populating all nodes in the graph.

**Sequence formulation with pointers.** To learn these conditionals efficiently, we serialize pointer-structured data as an interleaved sequence of *pointer* and *content* tokens $(p_0, v_0, p_1, v_1, \dots)$ and train a causal autoregressive transformer:

$$\Psi(X,p) \equiv \Pr\big[v_k \,\big|\, p_0, v_0, \dots, p_{k-1}, v_{k-1}, p_k\big].$$

Pointer tokens remove raster-order bias and enable *random access*: the next pointer can be predicted or provided as conditioning. Thus, decoding order itself becomes a controllable traversal of PGM nodes, supporting fully sequential, fully parallel, or hybrid schedules, while keeping a single next-token objective. This recasts learning a high-dimensional PGM as standard GPT-style training on many random traversals.

**Strict locality via HLQ.** Instead of compressing an entire frame into global codes, we use a Hierarchical Local Quantizer (HLQ), a convolutional autoencoder whose receptive field is restricted to each patch, ensuring *strict* locality during encoding. Each patch is encoded into a short sequence of four codes, where the first provides a coarse preview and the remaining codes progressively add fine detail. This property ensures that local interventions such as masking, overwriting, or resampling individual patches behave in a predictable way. Moreover, the resulting code structure makes the autoregressive modeling objective better aligned with natural language modeling assumptions of conditional independence, enabling tractable factorization and controllable local edits (see Figure 1).

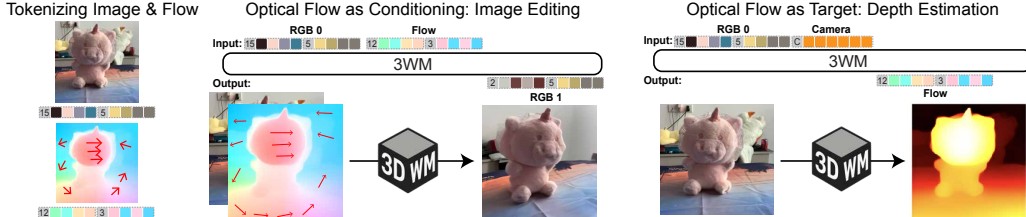

**Figure 2: Flexible inference pathways across modalities.** Our framework allows us to flexibly construct inference pathways for 3D scene understanding. Using optical flow tokens as conditioning, the model performs image editing by generating the next RGB frame. Conversely, when optical flow tokens serve as the prediction target, the model enables depth estimation by predicting the next flow field from a single RGB image and in-plane camera motion input.

We evaluate the contribution of the local random access sequence design through an ablation study provided in Appendix A.4. Further details on datasets, filtering, and training of 3WM are provided in Appendix A.1.

## 3.2 FLEXIBLE INFERENCE PATHWAYS

**Optical flow as a control surface.** Optical-flow patches serve as a powerful control mechanism within our PGM, providing an explicit representation of motion that users can directly manipulate. We adopt the causal ordering $[\text{RGB}, C] \rightarrow \text{Flow} \rightarrow \text{RGB}$ in our sequences, where flow acts as an intermediate action space: each patch specifies *what moves* and *by how much* at that spatial location. This design enables users to provide sparse or dense flow constraints to guide generation, with appearance synthesis and disocclusion resolution emerging naturally from the RGB predictions that follow.

With this flexible design, we realize several inference pathways using the same model:

- $\Psi(\text{RGB}_0, F_{0 \rightarrow 1}) \rightarrow \text{RGB}_1$: Dense flow-controlled RGB prediction—given an input image and a pre-defined dense flow field (motion instructions), render the next frame.

- $\Psi(\text{RGB}_0, F_{\text{sparse}}) \rightarrow F_{0 \rightarrow 1}$: Sparse-to-dense flow completion—given an input image and sparse flow seeds, complete a dense flow field. Can be composed with dense flow-controlled RGB prediction to enable object removal and localized edits.

- $\Psi(\text{RGB}_0, C_{\text{in-plane}}) \rightarrow F_{0 \rightarrow 1}$: Camera-controlled flow prediction—given an input image and a camera translation, obtain the globally induced flow field. Can be utilized for depth estimation via induced parallax.

Each inference pathway represents a different conditional query $\Psi(X, p)$ over the same underlying joint distribution, where we condition on different subsets of nodes (RGB, flow, or camera) and sample the remaining. The task itself is defined entirely by which nodes we choose to observe versus predict: dense flow conditioning yields motion control, sparse flow yields completion, and camera conditioning yields structure-from-motion. This PGM formulation eliminates the need for task-specific architectures; instead, diverse capabilities emerge as different paths through the graph. The effectiveness of optical flow as an intermediate representation is validated in Appendix A.4.

## 3.3 ZERO-SHOT PROMPTS FOR 3D TASKS

**Novel view synthesis** Novel view synthesis can be performed with the $\Psi(\text{RGB}_0, F_{0 \rightarrow 1}) \rightarrow \text{RGB}_1$ inference pathway, by conditioning the model on 2D optical flow fields that represent how the pixels move given a desired camera pose change. To generate these flow fields, we use the following steps: a) estimate depth from the input image using an off-the-shelf model Yang et al. (2024); Wang et al. (2024a), and unproject it into a 3D point cloud, b) apply a rigid transformation to the point cloud given camera transformations, c) re-project the transformed point cloud and compute the displacement relative to the pixels of the first frame to obtain the 2D flow (See Figure 2 and Appendix A.2). Finally, 3WM generates the edited image conditioned on the computed flow map and the input image.

**3D object manipulation** 3D object manipulation can be performed with the $\Psi(\text{RGB}_0, F_{0 \rightarrow 1}) \rightarrow \text{RGB}_1$ inference pathway by creating a flow field where the flow on the surface of the object charac-

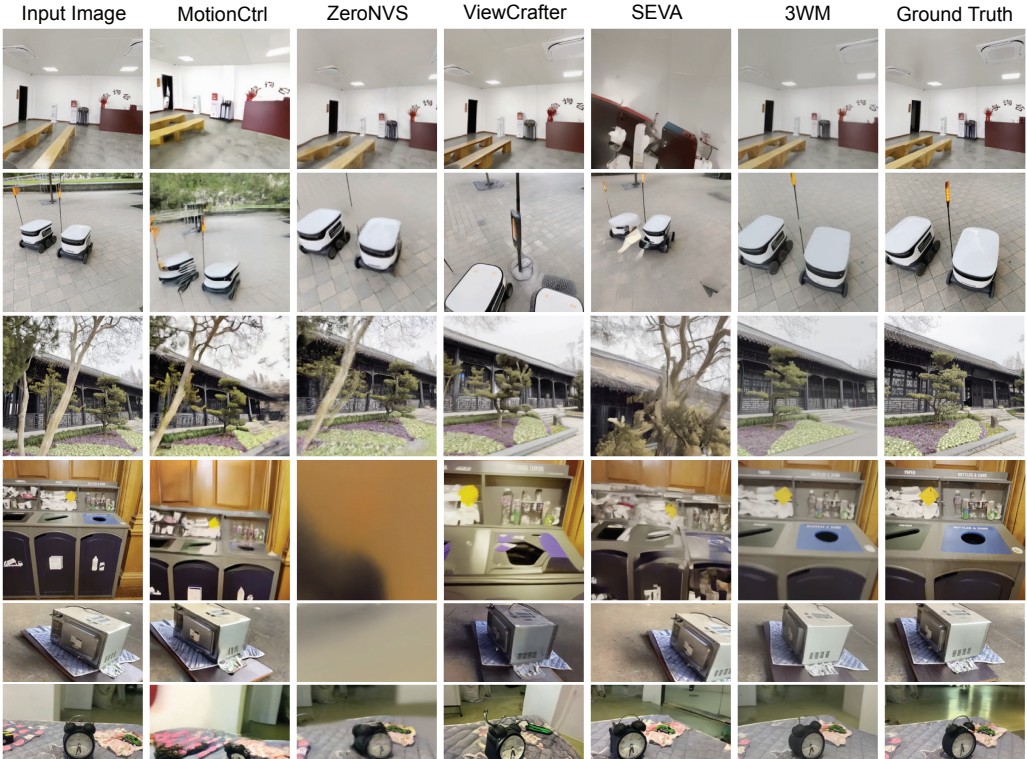

**Figure 3: Novel view synthesis from a single image.** The results show that our model performs controllable novel view synthesis with various camera motions in a diverse scenes. Compared to other models, the reconstructed images do not show abrupt change in object and scene identity.

terizes the 3D transformation to be performed, with the flow of the background set to 0 – conditioning the predictor to move the object, but keep the background fixed. We follow a similar procedure described above to produce flow fields for rigid object transformations and use the SegmentAnything Kirillov et al. (2023) model to suppress the flow of the background regions (See Figure 2 and Appendix A.2). Additionally, we can use the $\Psi(\text{RGB}_0, F_{\text{sparse}}) \rightarrow F_{0 \rightarrow 1}$ pathway to produce dense flow fields corresponding to object motion from a sparse flow prompt.

**Depth extraction** Camera-conditioned flow generation through the $\Psi(\text{RGB}_0, C_{\text{in-plane}}) \rightarrow F_{0 \rightarrow 1}$ inference pathway provides a natural method for extracting depth maps without additional finetuning. We provide in-plane camera motion as input to 3WM and predict the optical flow induced by camera motion. Then, we compute the magnitude of the optical flow to compute the disparity which, when inverted, yields 2.5D depth maps. That is, $D_{\text{depth}} \propto \frac{1}{F_{\text{flow}}}$, where $F_{\text{flow}} = \Psi(RGB, C_{\text{in-plane}})$. In practice, we find that a simple downward camera translation is sufficient to generate high-quality depth maps. Additionally, performance can be improved by statistical aggregation over disparity maps generated with different seeds for the same image.

## 4 RESULTS

We begin by evaluating novel view synthesis from a single image, demonstrating that 3WM outperforms prior NVS models across object-centric and scene-level benchmarks. We then present results on 3D object manipulation, using our 3DEditBench dataset to compare edit fidelity and geometric adherence against diffusion- and drag-based baselines. Lastly, we examine self-supervised depth estimation, showing that 3WM achieves strong performance on both static and dynamic indoor datasets, despite never using depth supervision during training.

### 4.1 NOVEL VIEW SYNTHESIS

**Evaluation Details.** We evaluate novel view synthesis (NVS) from single images on two out-of-distribution benchmarks: WildRGB-D for object-centric NVS and DL3DV for scene-level NVS. We also report results on the SEVA benchmark for single-image NVS (small-viewpoint, Reconfusion

| Model | Our Evaluations | | | | SEVA Benchmark | | | | | |
|---|---|---|---|---|---|---|---|---|---|---|
| | WildRGB-D | | DL3DV | | RE10K | | LLFF | | DTU | |
| | PSNR ↑ | LPIPS ↓ | PSNR ↑ | LPIPS ↓ | PSNR ↑ | LPIPS ↓ | PSNR ↑ | LPIPS ↓ | PSNR ↑ | LPIPS ↓ |
| MotionCtrl | 12.39 | 0.404 | 12.62 | 0.462 | - | - | - | - | - | - |
| ZeroNVS | 16.14 | 0.283 | 15.62 | 0.331 | - | - | - | - | - | - |
| ViewCrafter | 13.96 | 0.290 | 16.59 | **0.253** | 20.88 | 0.287 | 10.53 | 0.620 | 12.66 | 0.485 |
| SEVA | 15.10 | 0.278 | 11.82 | 0.516 | 18.11 | 0.308 | 14.03 | **0.389** | **14.47** | **0.316** |
| **3WM (Ours)** | **18.02** | **0.185** | **19.02** | 0.252 | **21.54** | **0.231** | **15.24** | 0.490 | 14.63 | 0.357 |

Table 1: **Comparison of metrics for novel view synthesis.** The left block reports results on WildRGB-D and DL3DV from our evaluation set. The right block presents SEVA benchmark Zhou et al. (2025) performance on the small-viewpoint NVS setting using the Reconfusion split across DTU, LLFF, and RE10K datasets.

split). Quantitative metrics include PSNR and LPIPS Zhang et al. (2018). We compare against MotionCtrl, ZeroNVS, ViewCrafter, and SEVA as baselines. Further details on dataset selection and implementation are provided in the appendix A.3.

**Qualitative and Quantitative Comparisons.** As shown in Table 1, our model achieves the best overall performance on WildRGB-D, DL3DV, and the SEVA benchmark, reflecting both reconstruction quality and precise camera control. Qualitatively (Figure 3), MotionCtrl distorts scenes and objects inconsistently, and despite efforts to optimize scene scales, it fails to accurately control camera motion. ZeroNVS often produces inaccurate 3D reconstructions with artifacts, and its hallucinated regions are blurry and unrealistic. ViewCrafter generates visually appealing images but frequently changes object appearance and global illumination. SEVA performs reasonably on WildRGB-D but requires scale sweeps for alignment and breaks down on DL3DV with more diverse camera trajectories. In contrast, 3WM maintains both object and scene identity while offering robust and reliable camera control.

## 4.2 3D Object Manipulation

| Model | PSNR ↑ | LPIPS ↓ | EA ↑ |
|---|---|---|---|
| DragAnything | 15.13 | 0.443 | 0.517 |
| Diffusion Handles | 17.82 | 0.344 | 0.619 |
| LightningDrag | 19.52 | 0.184 | 0.722 |
| **3WM (ours)** | **22.73** | **0.133** | **0.797** |

**Table 2: Comparison of metrics for 3D object manipulation.**

**Evaluation Details.** We compare our method against DiffusionHandles Pandey et al. (2024), the closest related work that performs 3D object edits using depth-conditioned diffusion models, as well as drag-based image editing approaches such as LightningDrag Shi et al. (2024) and DragAnything Wu et al. (2024). While the latter cannot be directly conditioned on 3D transforms, we adapt them by providing sparse 2D flow vectors from our dataset annotations, which enables them to perform 3D manipulations to a reasonable extent. For quantitative evaluation, we follow the metrics used in our NVS experiments, including PSNR and LPIPS. However, prior work Pandey et al. (2024) has shown that these metrics tend to favor perceptual image quality over edit accuracy. To address this, we also report the Edit Adherence (EA) metric, which measures how well the boundaries of the manipulated object align with the ground truth.

**New Object Editing Benchmark.** Most prior work in this area either use human evaluations on a small set of images Michel et al. (2023), or synthetic benchmarks Pandey et al. (2024) to evaluate their method. This can be attributed to the lack of high quality real-world datasets with ground-truth 3D object transform annotations. To address this problem, we collect a dataset called **3DEditBench** consisting of 100 image pairs with a diverse set of object types undergoing rotations and translations, and inter-object occlusions. We capture these images in a variety of background and lighting conditions. To obtain the ground-truth 3D object transformation for a given pair, we annotate four corresponding points in the two images, unproject them, and use least-squares opti-

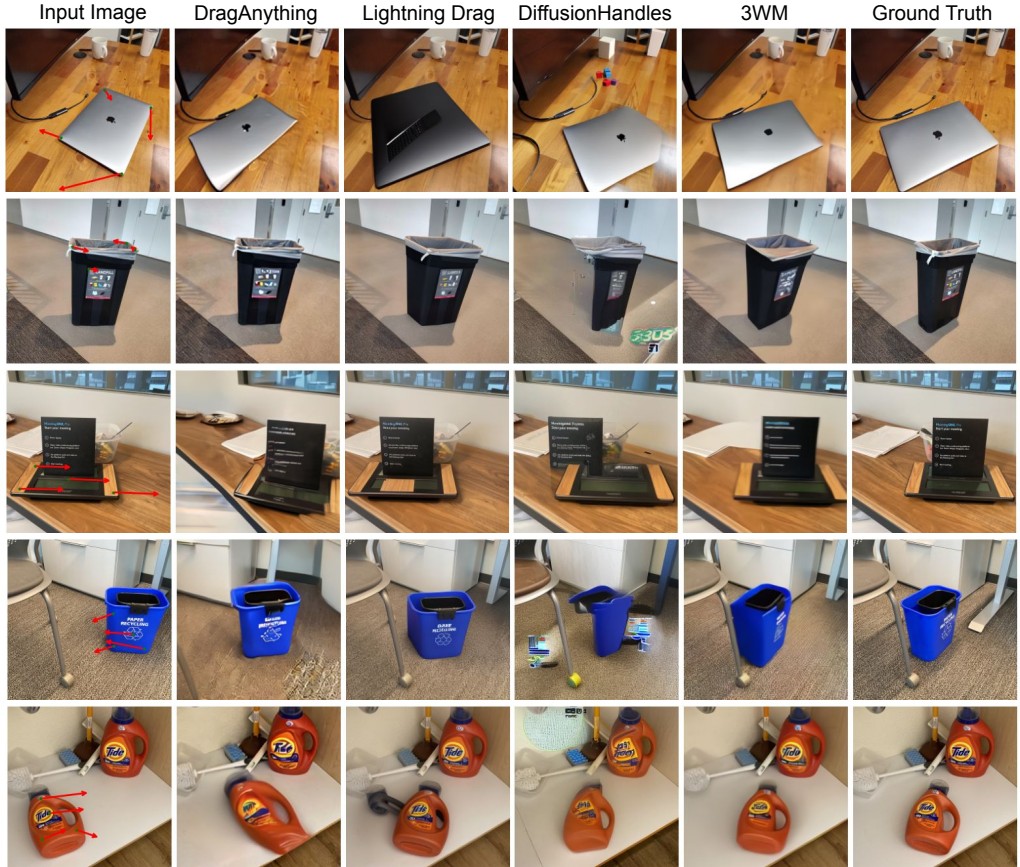

**Figure 4: 3D object manipulation from a single image.** We show that our model can perform both 3D object translation and rotation. Compared to other methods, our model preserves object identity on real world images, and produces more photorealistic generated images with accurate object edits.

mization to find the best-fitting rigid transformation that aligns the two sets of points. This transform is then used to create flow maps that condition 3WM to perform 3D object edits in natural scenes.

**Qualitative and Quantitative Comparisons.** We find that our model outperforms other methods on all metrics (see Table 2). Qualitative comparisons in Figure 4 further show that our model produces more accurate and realistic results, particularly on complex 3D transformations. The Edit Adherence (EA) metric Pandey et al. (2024), designed to measure how precisely the edited object matches the intended transformation, strongly favors our model's generations. Interestingly, DiffusionHandles Pandey et al. (2024) often struggles on real-world images due to failures in the null-text inversion process, which causes changes in surrounding object appearance and leads to unnatural generations, blurry reconstructions, and incorrect 3D motion. A similar trend is observed in the drag-based baselines, though LightningDrag degrades less severely. In contrast, 3WM overcomes these issues and produces geometrically consistent edits.

### 4.3 SELF-SUPERVISED DEPTH ESTIMATION

| Model | NYUD-v2 | | BONN | | TUM | |
|---|---|---|---|---|---|---|
| | AbsRel ↓ | $\delta_1$ ↑ | AbsRel ↓ | $\delta_1$ ↑ | AbsRel ↓ | $\delta_1$ ↑ |
| SC-DepthV2 | 0.132 | 0.828 | 0.172 | 0.814 | 0.257 | 0.582 |
| IndoorDepth | 0.116 | 0.864 | 0.154 | 0.846 | 0.205 | 0.697 |
| MotionCtrl | 0.232 | 0.664 | 0.171 | 0.793 | 0.205 | 0.682 |
| SEVA | 0.404 | 0.574 | 0.352 | 0.618 | 0.503 | 0.496 |
| **3WM (Ours)** | **0.078** | **0.940** | **0.084** | **0.942** | **0.137** | **0.869** |

**Table 3: Comparison of metrics for self-supervised monocular depth estimation on NYUD-v2, BONN, and TUM datasets.**

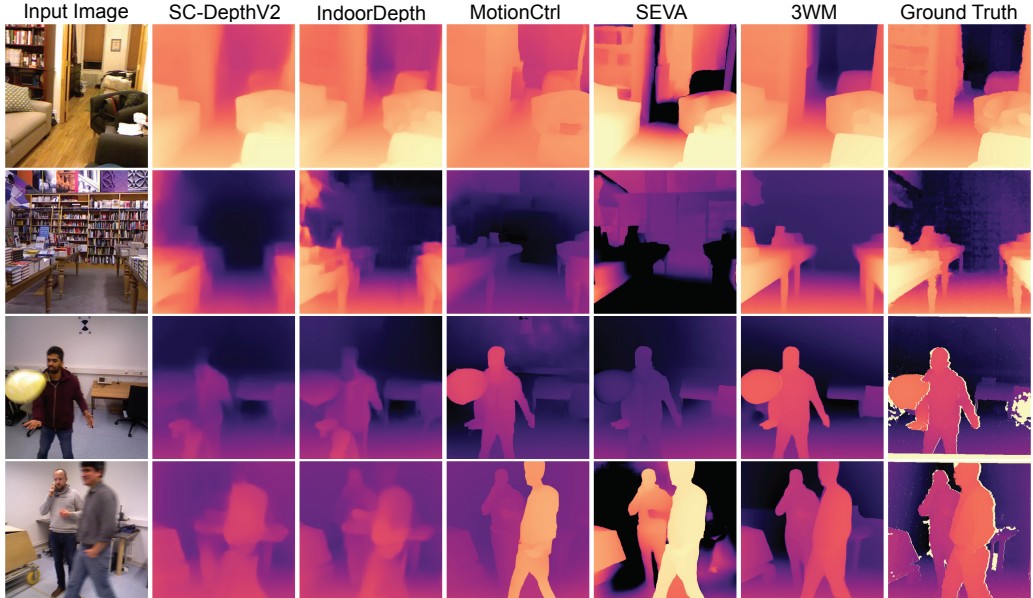

**Figure 5: Self-supervised monocular depth estimation.** On both static and dynamic scenes, our model outperforms existing self-supervised depth estimation methods. Compared to other generative model baselines, it shows stronger camera motion controllability and geometric understanding. Yellow artifacts in ground truth depth maps are noise and excluded during the evaluation.

**Evaluation Details.** Since we do not use any depth label during the training, for a fair comparison, we compare our model with other self-supervised depth estimators. We evaluate SC-DepthV2 Bian et al. (2021), IndoorDepth Fan et al. (2023), MotionCtrl Wang et al. (2024c), and SEVA Zhou et al. (2025) as baselines, aligning with the self-supervised setting of our model. To extract depth from MotionCtrl and SEVA, we induce a downward in-plane camera motion and compute the disparity between the first and generated images using DPFlow Morimitsu et al. (2025). We evaluate the performance on three datasets: NYUv2 Silberman et al. (2012), BONN Palazzolo et al. (2019), and TUM Sturm et al. (2012). NYUv2 is mostly composed of static scenes, whereas BONN and TUM include humans with implied motion. To ensure consistent input resolution across methods, we evaluate only the center-cropped regions of each image. Unlike relative depth estimation methods that apply scale-and-shift alignment during evaluation Ranftl et al. (2020), we adopt only a global scale adjustment using the median depth value.

**Qualitative and Quantitative Comparisons.** As shown in Table 3 and Figure 5, 3WM achieves high-quality depth reconstruction in both static and dynamic settings, outperforming all baselines across diverse evaluations. The self-supervised baseline models exhibit limitations in dynamic scenes because they rely on static geometry consistency, preventing them from extracting training signals from moving objects. In contrast, our model effectively learns depth cues from optical flow in unconstrained video. Although MotionCtrl and SEVA yield qualitative improvements on dynamic objects compared to other baselines, their overall performance remains weak, indicating limited controllability and geometric understanding. These results demonstrate the strong geometric understanding and controllability of our model compared to existing baselines.

## 5 EMERGENT GEOMETRIC REASONING ABILITIES

Geometric reasoning in 3D scenes is inherently complex, requiring the ability to address scenarios that extend beyond any single benchmark task. These include uncovering occluded geometry, navigating cluttered environments, and handling ambiguous depth.

Figure 6 illustrates several representative cases. (a) 3WM can first perform object manipulation to move obstacles aside and then apply novel view synthesis to the updated scene. This sequential reasoning exposes free space that was previously occluded, allowing navigation through newly opened paths. (b) When the occluding object cannot be physically moved, the model should be able to simulate navigation along complex trajectories to reveal hidden regions. In such a case, 3WM can

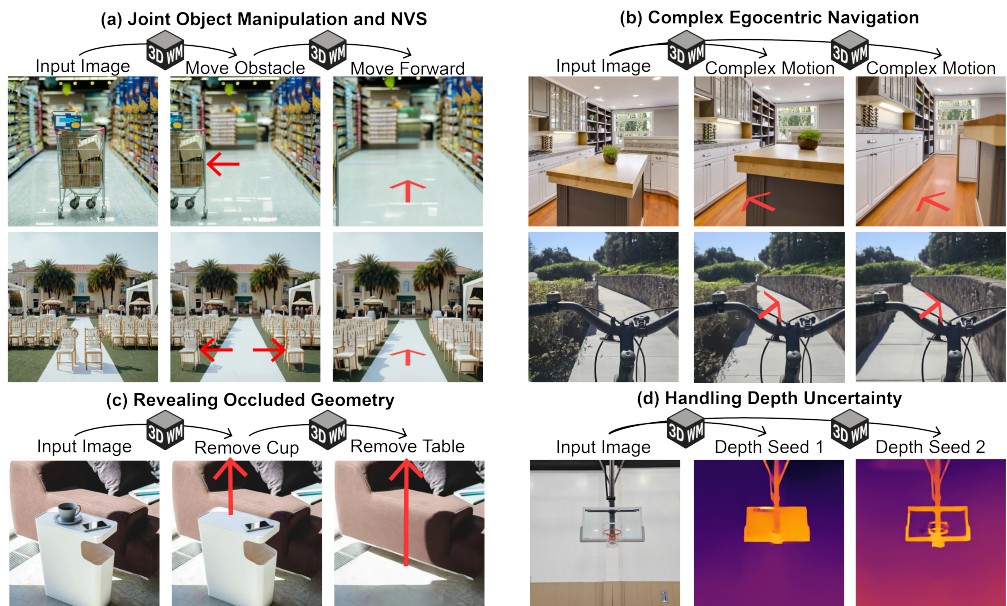

Figure 6: **Flexible geometric reasoning in the complex real-world environment through 3WM.**
(a) The model moves obstacles aside to reveal free space and simulates navigation through the newly opened path. (b) The model follows complex egocentric trajectories to uncover hidden regions and moves together with objects to capture realistic navigation scenarios. (c) The model removes attached objects one by one to reveal the background geometry, performing a form of amodal completion. (d) The model handles depth uncertainty in cases such as transparent objects by generating multiple plausible depth outputs.

move forward, lower the viewpoint, and rotate the camera to the right to expose a hallway. The model can also support navigation with objects, as in moving with a bike, where the bike remains fixed relative to the egocentric viewpoint while the background updates with the camera motion. (c) To reveal geometry hidden by attached objects, the model applies a large flow displacement to the object targeted for removal and repeats this process iteratively. This allows the model to uncover both the background and the previously occluded surfaces of other objects, effectively performing amodal completion. (d) When conditioned on an in-plane camera motion, the model can generate multiple optical flow maps corresponding to different plausible depth configurations. This captures multimodal depth reasoning, which is essential in cases such as transparent or reflective materials where depth cannot be uniquely defined.

These examples demonstrate that our model is not limited to predefined tasks or narrow benchmarks. By supporting flexible inference pathways, it can generalize to a wide range of challenges that arise in real-world 3D environments. This ability to reason flexibly about geometry highlights the strength of our framework and points toward general-purpose models for 3D understanding.

## 6    DISCUSSION

We present several representative failure cases in Fig. 11. These behaviors occur occasionally rather than systematically, and are already reflected in our quantitative evaluation, where our model still outperforms prior methods. The observed failure modes include motion-induced blur under large displacements, residual object copies at the original location, and sensitivity to segmentation quality.

## 7    CONCLUSION

In this work, we demonstrate that 3WM can unify novel view synthesis, object manipulation, and depth estimation through diverse inference pathways while maintaining precise controllability and strong geometric consistency. By showing that complex 3D reasoning can emerge naturally within a unified system, 3WM provides a step toward general-purpose visual world models capable of supporting richer interaction with the physical environment.

ACKNOWLEDGMENTS

This work was supported by the following awards: To D.L.K.Y.: Simons Foundation grant 543061, National Science Foundation CAREER grant 1844724, National Science Foundation Grant NCS-FR 2123963, Office of Naval Research grant S5122, ONR MURI00010802, ONR MURI S5847, and ONR MURI 1141386- 493027. We also thank the Stanford HAI, Stanford Data Sciences and the Marlowe team, and the Google TPU Research Cloud team for computing support.

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

# A APPENDIX

## A.1 DATASETS AND TRAINING DETAILS

**Training Datasets.** 3WM was pre-trained on a combination of a large-scale internet video collection, termed `BVD` (Big Video Dataset), and several 3D vision benchmarks. The latter include the train splits of ScanNet++ Yeshwanth et al. (2023), CO3D Reizenstein et al. (2021), RealEstate10K Zhou et al. (2018), MVImgNet Yu et al. (2023), DL3DV Ling et al. (2024), and EgoExo4D Grauman et al. (2024). Optical flow was computed for all videos using DPFlow Morimitsu et al. (2025) and UFM Zhang et al. (2025).

**Big Video Dataset.** The `BVD` consists of roughly 7,000 hours of internet videos automatically crawled using search queries generated by LLaMA 3 Grattafiori et al. (2024). The queries targeted videos containing rich physical dynamics, diverse environments, and varied objects. Specifically, action categories from Kinetics400 Kay et al. (2017) were expanded with additional sports, physical activities, and product review categories. To ensure training relevance, we filtered videos by requiring a minimum level of optical flow and by applying CLIP Radford et al. (2021)-based keyword alignment. Positive keywords included *action*, *activity*, *motion*, and *place*, while negative keywords included *animation*, *cartoon*, *face*, *game menu*, *graphic*, *map*, *newscast*, *person*, and *screenshot*. Alignment was quantified by the dot product between CLIP embeddings of keywords and video frames.

**Training Details.** The 3WM sequence model was trained autoregressively using cross-entropy loss on next-token prediction with a batch size of $512$ and a sequence length of $4,096$. Training for RGB and camera pose tokens ran for 500K steps. The learning rate was linearly warmed up over 2K iterations to $3 \times 10^{-4}$, held constant until 500K steps. Training was then continued for an additional 200K steps with optical flow tokens, where the learning rate stayed at $3 \times 10^{-4}$ initially and decayed linearly to zero during the last 100K steps.

The RGB variant of HLQ was trained on a combination of ImageNet and OpenImages with a batch size of $512$ for 200K iterations. The objective combined an $\ell_1$ reconstruction loss, a low-resolution loss, and a DinoV2-based perceptual loss. Optimization was performed using AdamW with a learning rate of $1 \times 10^{-4}$, including 2K warmup steps followed by 198K steps of decay to zero.

The flow variant of HLQ was trained on the same dataset used for training the sequence model, with a batch size of $512$. Optical flow was extracted from video data using DPFlow. Optimization was again performed using AdamW with a learning rate of $1 \times 10^{-4}$. Training used 2K warmup steps, followed by 300K iterations with a fixed learning rate and then 200K iterations with linear decay.

**Training Sequence.** Every training example is formed as a causal token sequence in which all previous tokens serve as observed context and the next token is supervised, identical to standard language-model training. In practice, all RGB and flow tokens after the first frame are predicted autoregressively. The only exceptions are the $\text{RGB}_0$ tokens, camera-pose tokens, and pointer tokens, which we do not supervise in this work, though the framework is not limited to this choice.

Within RGB+flow sequences, the model naturally encounters several supervision patterns depending on whether flow is the final target or an intermediate variable used to produce the next RGB frame. Some sequences terminate with flow (e.g., $\text{RGB}_0 \to \text{Flow}_{0\to1}$ or $\text{RGB}_0 \to \text{RGB}_1 \to \text{Flow}_{1\to2}$), while others place flow mid-sequence and continue autoregressively to the next RGB frame (e.g., $\text{RGB}_0 \to \text{Flow}_{0\to1} \to \text{RGB}_1$ or $\text{RGB}_0 \to \text{RGB}_1 \to \text{Flow}_{1\to2} \to \text{RGB}_2$). When available, pose tokens are inserted between RGB frames or immediately before flow. These variations provide balanced supervision across modalities and naturally expose the model to different geometric reasoning configurations.

The curriculum over training steps controls which of these sequence types the model sees. Before 500k steps, training uses RGB-only sequences drawn from 2–4 consecutive frames, where each frame is partially masked according to fixed visible-token ratios ($[0.5, 0.3]$ for 2-frame sequences, $[0.5, 0.15, 0.15]$ for 3-frame sequences, and $[0.5, 0.1, 0.1, 0.1]$ for 4-frame sequences). After 500k steps, we introduce flow and train on a mixture of RGB-only and RGB+flow sequences, with random allocation of tokens across frames, modalities, and time indices. This particular schedule simply reflects one reasonable choice rather than an optimized design. The same autoregressive formulation

supports arbitrary modality mixtures, token allocations, and sequence orderings, and exploring alternative training schedules and sequence constructions would be an interesting direction.

## A.2   3D EDITING METHODS

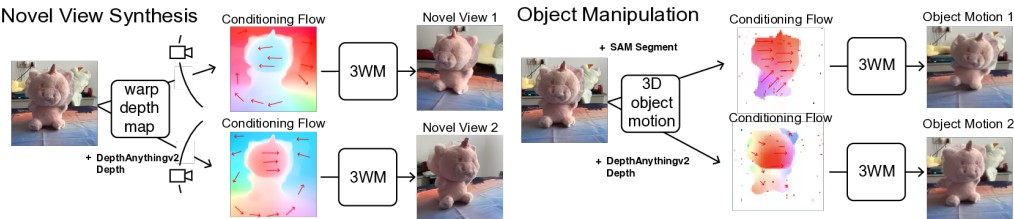

**Figure 7: 3D scene editing through optical flow field manipulation:** We perform 3D scene edits by constructing optical flow fields corresponding to the desired transformations - either camera or object motion in 3D.

We illustrate the procedure for constructing flow fields used in both novel view synthesis and 3D object manipulation. For novel view synthesis (Section 3.3), the flow encodes pixel displacements induced by a desired camera motion. For object manipulation (Section 3.3), the flow specifies the motion of object surfaces under a 3D transformation while background regions remain fixed. Figure 7 provides a unified visualization of these processes, complementing the method descriptions in the corresponding subsections.

## A.3   NVS EVALUATION DETAILS

To ensure a fair evaluation of novel view synthesis (NVS) on out-of-distribution datasets, we selected two benchmarks: WildRGB-D for object-centric NVS and DL3DV for scene-level NVS. For WildRGB-D, we randomly sampled 100 scenes from its evaluation split. For DL3DV, since some models were trained on this dataset, we selected 100 scenes from its recently released 11K subset, which, to the best of our knowledge, was not used to train any of the compared models. From each video, we extracted a 25-frame sequence and used the first frame as the input image and evaluated the generated frames. For quantitative evaluation, we measured PSNR and LPIPS Zhang et al. (2018). As baselines, we compared against MotionCtrl, ZeroNVS, ViewCrafter, and SEVA.

To evaluate novel view synthesis, we compared generated images to ground-truth real-world images using known camera poses. While camera rotation is unambiguous, camera translation may have arbitrary scale. Therefore, it was necessary to find the right scene scale to perform fair evaluations for all of the models. To align MotionCtrl, ZeroNVS, and SEVA results with ground-truth images, we swept a range of scene scales and took the generated trajectories with the best median LPIPS score across frames. For ZeroNVS, we swept scales in the range 0.1 to 10, multiplying the scale by the ground-truth camera translations from each evaluation scene. ZeroNVS introduces a normalization scheme Sargent et al. (2024) at training time to address this scale ambiguity, but does not apply it at inference. For MotionCtrl, we swept the range 1 to 10, as smaller translation scales empirically weaken the camera conditioning and lead to incorrect camera pose trajectories. For SEVA, we swept the scale from 0.1 to 2.0 and used their set NVS method for evaluation. Scale alignment for these models often failed for samples with especially poor 3D reconstruction quality. For ViewCrafter, we resolved the scene scale using their method of aligning point clouds from DUSt3R Wang et al. (2024a). For 3WM, we computed a single scale value per scene by matching the optical flow computed from the video using DPFlow and the 2D flow computed using the depth from DepthAnythingV2 or DUSt3R and relative camera pose changes.

Since ViewCrafter operates on wide rectangular videos, we adapted the input images accordingly. For DL3DV, which consists of wide images, we provided the full image to ViewCrafter. For WildRGB-D, whose images have greater height than width, we cropped them into wide rectangular regions to match ViewCrafter's aspect ratio. All other models received a center-cropped square image as input for both datasets. All evaluation metrics were computed only on the overlapping regions. For WildRGB-D, this region was rectangular, and for DL3DV it was square.

## A.4 ABLATION STUDIES

We conduct several ablation studies to validate two key design choices of our framework: the use of local random access sequence modeling and the use of optical flow as an intermediate representation.

**Local random access sequence modeling.** We compare different tokenization and sequence modeling strategies using 100M-parameter models in Table 4. Our approach with local random sequence shows clear benefits over the raster order approach. As observed in MAE He et al. (2022), VideoMAE Tong et al. (2022), and CWM Bear et al. (2023), random masking encourages stronger representation learning while allowing us to represent each frame with fewer tokens. In contrast, raster order models must encode all patches sequentially, leading to inefficiency and degraded performance. Moreover, we find that local tokens provide finer control over scene elements than global token alternatives Esser et al. (2021); Rombach et al. (2022), improving both controllability and output quality. Together, these results demonstrate that local random access sequence design not only improves training efficiency but also enhances the model's ability to support robust controllability in flexible inference pathways.

**WildRGB-D: Novel View Synthesis**

| Model (100M) | PSNR ↑ | SSIM ↑ | LPIPS ↓ |
|---|---|---|---|
| **Local & Random** | **17.28** | **0.530** | **0.236** |
| **Local & Raster** | 15.00 | 0.459 | 0.385 |
| **VQGAN & Random** | 17.16 | 0.515 | 0.238 |
| **VQGAN & Raster** | 15.71 | 0.454 | 0.298 |

Table 4: **Advantage of local random access sequence modeling.** Comparison of 100M models with different tokenizers and sequence strategies shows the benefit of random access. Local tokens further improve controllability of the scene, yielding the best overall performance.

**Optical flow as causal intermediate.** Table 5 shows that using optical flow as a control signal substantially improves performance across tasks. Flow provides a direct handle on scene geometry, enabling more precise inference than camera-only control, which suffers from scale ambiguity. This advantage is most apparent in depth estimation, where predicting flow directly yields more geometrically consistent reconstructions than first predicting RGB and deriving flow afterward from the input image and predicted image. These results support our claim that intermediate modalities such as flow enable more controllable and robust inference pathways, grounding the model in physical structure.

**WildRGB-D: Novel View Synthesis**

| Model | PSNR ↑ | SSIM ↑ | LPIPS ↓ |
|---|---|---|---|
| **3WM$_{rgb}$** | 14.49 | 0.389 | 0.346 |
| **3WM** | **18.02** | **0.555** | **0.185** |

**NYU Depth Estimation**

| Dataset | AbsRel ↓ | Log10 ↓ | $\delta_1$ ↑ |
|---|---|---|---|
| **3WM$_{rgb}$** | 0.173 | 0.064 | 0.825 |
| **3WM** | **0.078** | **0.033** | **0.940** |

Table 5: **Advantage of optical flow for causal inference.** The results highlight the advantage of optical flow as a control signal. Optical flow provides a direct handle on scene geometry, enabling more precise inference than camera-only control, which is affected by scale ambiguity. This benefit is also reflected in depth estimation, where predicting flow directly outperforms deriving it from predicted image.

## A.5 LIGHTING AND APPEARANCE UNDERSTANDING

Our model does not yet capture lighting effects perfectly across all conditions. At the same time, we consistently observe many cases where the model correctly predicts shadows and view-dependent lighting changes, indicating that it learns meaningful aspects of object appearance (Figure 8). We do not regard the remaining failure cases in challenging lighting as fundamental limitations of the formulation. Rather, they reflect the current scale of the model and the limited diversity of illumina-

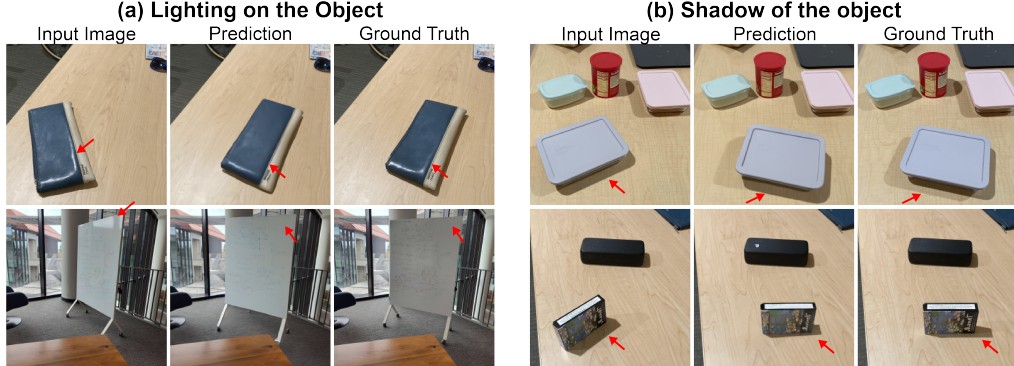

**Figure 8: Lighting and appearance understanding.** Additional qualitative examples illustrating the model's handling of lighting and appearance. In case (a), specular highlights on objects change appropriately as the object moves, and in case (b), cast shadows shift consistently with the object's motion. While some examples still show incomplete specular or shading behavior, many exhibit correct reasoning about shadows and view-dependent appearance. These results suggest that lighting fidelity is primarily constrained by model and data scale rather than by limitations of the approach.

tion conditions in the training data. We expect lighting fidelity to improve as the model is scaled up and trained on datasets with broader lighting variation.

### A.6 EXTENDED DISCUSSION ON OBJECT MANIPULATION

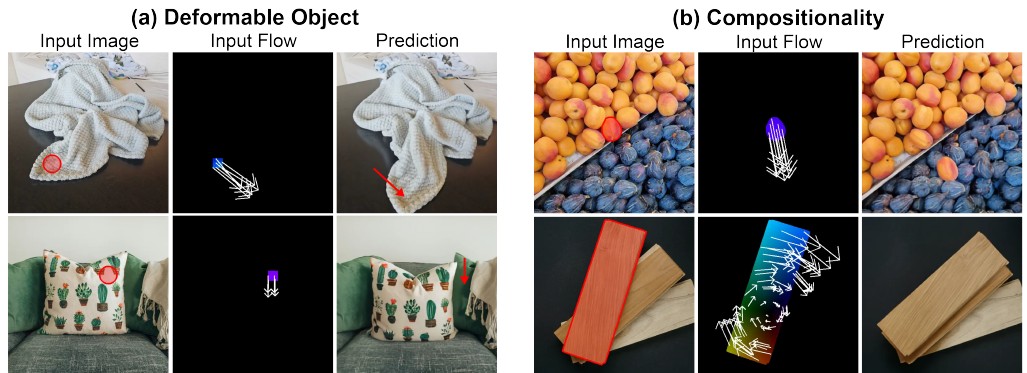

**Figure 9: Additional qualitative examples of object manipulation.** (a) The model performs deformable-object manipulation by applying flow conditioning only to the deforming regions, which allows it to infer the resulting non-rigid transformations. In the figure, the red circles indicate regions where non-zero flow is applied on the object. Other regions are left unspecified, requiring the model to infer their behavior from the local conditioning. (b) In multi-object scenes, the model achieves selective manipulation by applying flow conditioning to a segmented target object while assigning zero flow to the rest of the scene.

The examples in the main paper focus on rigid object manipulation, reflecting the fact that **3DEditBench** primarily contains rigid transformations. However, our framework is not limited to rigid object manipulation. Because the model operates on local tokens and the conditioning interface supports both sparse and dense flow control, it can naturally handle deformable object manipulation. By applying flow conditioning only to the deforming region and leaving the rest of the object unconstrained, the model infers the resulting non-rigid transformation. We additionally set the flow to zero in the four corners of the image to signal that the transformation is local rather than global. We have added qualitative examples that illustrate this behavior in Figure 9. Finally, we show examples with and without stop patches on the blanket to illustrate how these signals affect the model's probabilistic behavior in Figure 10.

For stacked or multi-object scenes, we include additional examples in Figure 9 where the model manipulates a single object while preserving the geometry and appearance of the others. In these cases, we segment the target object and apply full flow conditioning to that object, while the rest

of the scene receives zero flow in the conditioning. These results show that the model can selectively operate on one object among several and maintain coherent scene structure throughout the manipulation.

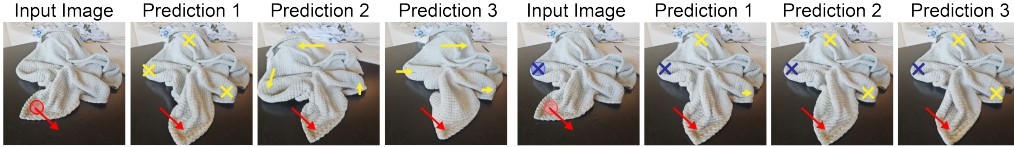

**Figure 10: Effect of stop patches on probabilistic deformation.** Left: predictions from three random seeds using only the motion patch (red). Right: predictions using both the motion patch (red) and stop patches (blue). Yellow arrow indicates the model's predicted deformation. Without stop patches, the model exhibits uncertainty in how the rest of the blanket moves. Stop patches constrain the motion and lead to more stable outcomes.

## A.7 AMODAL COMPLETION EVALUATION

| Model | AbsRel ↓ | Log10 ↓ | $\delta_1$ ↑ |
|---|---|---|---|
| DragAnything | 0.0666 | 0.0285 | 0.9480 |
| Diffusion Handles | 0.0500 | 0.0215 | 0.9680 |
| Lightning Drag | 0.0324 | 0.0145 | **0.9782** |
| **3WM (Ours)** | **0.0263** | **0.0120** | 0.9740 |

**Table 6: Amodal completion depth evaluation on `3DEditBench`.** Metrics are computed on the region originally occupied by the moved object.

Here, we show a quantitative amodal completion evaluation. Since all baseline editing methods in the paper can perform amodal completion by moving objects, we assess how accurately each method reconstructs the depth behind the removed object on 3DEditBench. We segment the region previously occupied by the object and compare the predicted depth in the edited image with the ground-truth depth of the target view, restricted to this occluded region. Depth maps are estimated using DepthAnything v2 and aligned by median-based scale normalization.

This metric directly measures a model's ability to infer the hidden geometry. As shown in Table 6, our model achieves lower AbsRel and Log10 errors than all baselines while maintaining competitive $\delta_1$, indicating more accurate reconstruction of the unseen surfaces.

## A.8 LIMITATIONS & FUTURE WORK

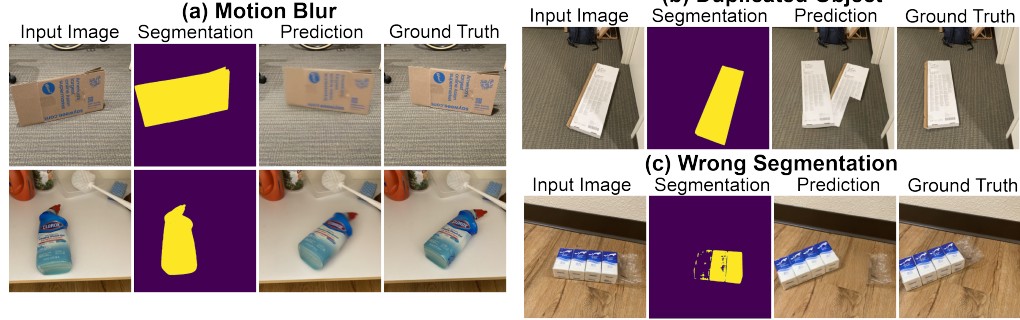

**Figure 11: Additional qualitative examples illustrating current limitations.** (a) Motion blur: Because the model is trained on real videos, it sometimes reproduces motion-induced blur when large object displacements are present. This behavior is consistent with the training distribution but may be undesirable for fine-grained manipulation. (b) Object duplication: The model may occasionally generate a duplicated copy at the original location. (c) Segmentation errors: For rigid-object manipulation, incorrect input segmentation leads to incorrect zero-flow constraints, causing unpredictable or distorted outcomes.

**Limitations**  As shown in Fig. 11, our method exhibits three failure modes: motion-induced blur under large displacements, residual object copies at the original location, and sensitivity to segmentation quality. These occur occasionally rather than systematically and are already captured in our quantitative results. Beyond visual artifacts, efficiency remains a practical constraint. Our runtime falls within the typical range for generative models but is not real-time, which limits use in interactive applications.

**Future Work**  We expect efficiency improvements from standard engineering optimizations widely used in large autoregressive models. Another promising direction is to evaluate the model's navigation and planning capabilities. We believe that exploring these directions would be highly valuable and represents natural next steps for unified physical world models.

## A.9  USE OF LARGE LANGUAGE MODELS (LLMS)

We used LLMs as a general-purpose assistance tool for grammar checking and for suggesting alternative expressions or synonyms. LLMs did not play a role in developing the research ideas, designing experiments, or writing substantive content.

