# OpenReview forum: "Unified 3D Scene Understanding Through Physical World Modeling"
_ICLR.cc/2026/Conference — ICLR 2026 Poster_

### Official Review · Reviewer_WZBe · 2025-10-26

**Soundness:** 3
**Presentation:** 2
**Contribution:** 2
**Rating:** 4
**Confidence:** 2

**Summary:**

This paper proposes 3WM (3D World Model), a unified framework for 3D scene understanding and interaction based on a probabilistic graphical model (PGM) formulation. Unlike task-specific 3D models, 3WM encodes multimodal scene elements (RGB, optical flow, camera pose) as nodes and performs diverse tasks—such as novel view synthesis, depth estimation, and 3D object manipulation—as different inference pathways through the same graph.

**Strengths:**

1. The paper introduces a conceptually elegant unification of multiple 3D vision tasks under one probabilistic and autoregressive modeling framework.
2. The local random-access sequence modeling design and pointer-token formulation are interesting and potentially valuable for scalable multimodal 3D reasoning.
3. Both quantitative and qualitative experiments have been conducted in sufficient detail.

**Weaknesses:**

1. The presentation quality is inconsistent. The Introduction section fails to clearly and explicitly highlight the main contributions of the paper, making it difficult for readers to grasp the motivation and significance of the work.
2. A teaser figure should be included in the Introduction to provide an intuitive overview of the proposed framework and help readers quickly understand the key idea and workflow.
3. The paper lacks comprehensive ablation studies, which are necessary to fully evaluate the generalization ability and effectiveness of the proposed method.

**Questions:**

I am not very familiar with the research areas of Novel View Synthesis and World Model, so my current evaluation mainly focuses on the presentation quality and writing clarity of the paper. I will update my assessment after considering feedback and technical insights from other reviewers.

---

> ### Author Response · Authors · 2025-11-16
> **Response regarding ablation studies.**
>
> Thank you for the helpful comment about ablations -- we wanted to write to you early about this because there might be a lengthy followup to do during the rebuttal period.
>
> One thing first is that we wanted to draw your attention to the two ablations studies in we've performed already, evaluating key design choices of the method (perhaps you might have missed them since they are a little bit buried in the appendices):
>
> -- Appendix A.1 analyzes local random access sequence modeling by comparing 100M parameter models with different tokenizer and sequence strategies. The results in Table 4 show that local random access tokens provide stronger controllability and lead to the best overall performance.
>
> -- Appendix A.1 also evaluates the role of optical flow as an intermediate control signal. Table 5 shows that using optical flow yields more precise geometry-driven inference than camera-only control, which is affected by scale ambiguity. The table also shows that predicting flow directly improves depth estimation compared with deriving flow from a predicted RGB image.
>
> With this in mind, we wanted to ask: what ablations would you feel would most strengthen the paper?  We would be grateful to know which aspects you are thinking about so we can try to address them directly. (One thing to bear in mind that these ablations can be pretty expensive computationally, so we need to plan pretty carefully if we are to do one during the rebuttal.)  Thanks!

---

> > ### Comment · Reviewer_WZBe · 2025-11-21
> >
> > Thank you for the clarification regarding the ablation studies in the Appendix—they look fine. Sorry for the delayed response, as I needed some time to re-examine the paper. And I believe several additional issues should be addressed:
> >
> > 1. Since efficiency is crucial for Embodied AI systems, could you report the efficiency of your model across different tasks, compared to the respective baselines, when used as a unified model?
> > 2. The discussion of related work is somewhat insufficient. Several recent and advanced 3D multimodal LLMs—such as **Inst3D-LMM (CVPR 2025)** and **Video-3D LLM (CVPR 2025)**—are not covered. Both of them can also be used as a unified model across different 3D scene understanding tasks. A more comprehensive review in the main paper would further strengthen this work.
> > 3. I also agree that the paper should include a discussion of the model’s limitations, as this would better inform readers about the scope and boundaries of the contributions.

---

> > > ### Author Response · Authors · 2025-11-22
> > >
> > > **“The Introduction section fails to clearly and explicitly highlight the main contributions…”**
> > >
> > > We thank the reviewer for this helpful observation. We have revised the Introduction section to explicitly state the core contributions in a clear and structured way. The revised version now provides a more direct explanation of the significance of the proposed unified model.
> > >
> > > **“A teaser figure should be included…”**
> > >
> > > We appreciate the suggestion to include a teaser figure. We have added a candidate teaser figure to the appendix to gather your feedback. This version reflects an older iteration of the model and is shown only to illustrate the possible form of the teaser, not its final content. We would be grateful for your thoughts on whether a teaser of this type would strengthen the main paper or whether you had a different style or purpose in mind.
> > >
> > > **“could you report the efficiency of your model across different tasks…”**
> > >
> > > Thank you for raising this important point. We conducted additional measurements comparing our inference time with task-specific baselines under the same hardware conditions. All experiments were performed on NVIDIA A40 GPUs (46 GB) running on dual-socket AMD EPYC 7352 systems (48 physical cores, 96 threads).
> > >
> > > For novel view synthesis:
> > >
> > > - ViewCrafter: 240 s (25 frames)
> > > - MotionCtrl: 62 s (14 frames)
> > > - ZeroNVS: 143 min (NeRF distillation), 2 min rendering (25 frames)
> > > - SEVA: 10 min 47 s (30 frames)
> > > - Ours: 253 s (1 frame)
> > >
> > > For 3D object manipulation (1 frame):
> > >
> > > - LightningDrag: 7.5 s
> > > - DragAnything: 99 s
> > > - Diffusion Handles: 135 s
> > > - Ours: 253 s
> > >
> > > For depth estimation from a single image:
> > >
> > > - SC-DepthV2: 5 ms
> > > - IndoorDepth: 79 ms
> > > - MotionCtrl: 62 s
> > > - ZeroNVS: 143 min (no quantitative evaluation)
> > > - SEVA: 10 min 47 s
> > > - Ours: 140 s
> > >
> > > For generative tasks, our runtime is within the typical generative-model range. Importantly, we have not applied dedicated engineering optimizations as this is the first model that performs autoregressive generation of RGB, optical flow, and camera pose within a unified model. We expect standard efficiency techniques used in large autoregressive models can improve the efficiency without conceptual obstacles. For the depth estimation task, we are not aiming to match the efficiency of specialized regression systems. The task remains valuable because it reveals the model’s underlying geometric understanding. If a fast estimator is desired in practice, the geometric knowledge in our unified model can be distilled into a lightweight regression model.
> > >
> > >
> > > **“The discussion of related work is somewhat insufficient…”**
> > >
> > > Thank you for suggesting an expansion of the related work discussion. Your comment highlighted that “3D scene understanding” is used differently across communities, so we clarify the definition used in this paper.
> > >
> > > In our work, 3D scene understanding refers to inferring the physical 3D structure of a scene from 2D inputs, and understanding how the physical scene and its future hypothetical observations transform under changes such as camera motion or object motion in 3D. More specifically, we ground this notion of 3D scene understanding in the capabilities required for depth estimation, novel view synthesis, and object manipulation, since these tasks jointly capture the challenges of perceiving geometry, inferring unseen views, and reasoning about how scenes evolve under physical transformations.
> > >
> > > The multimodal 3D LLMs you mentioned, including Inst3D-LMM and Video-3D LLM, target a different domain. They operate on pre-reconstructed 3D inputs and address semantic tasks such as 3D dense captioning, 3D visual grounding, and 3D question answering. These systems interpret an already-built 3D scene and do not reason about how the physical scene is formed from images or how future observations would change under physical transformations.
> > >
> > > We have included these works in the revised related work section to clearly contrast them with our work.
> > >
> > > **“the paper should include a discussion of the model’s limitations…”**
> > >
> > > Thank you for the suggestion. We have added a dedicated “Limitations & Future Work” subsection in the appendix, including illustrative examples of occasional failure cases such as motion-induced blur, object duplication, and sensitivity to segmentation quality during rigid object manipulation. These behaviors occur infrequently and are already reflected in our quantitative metrics, where our model continues to outperform prior work.
> > >
> > >
> > > **List of change in the paper:**
> > >
> > > - Added an explicit summary of contributions in the introduction. (Introduction)
> > > - Included a candidate teaser figure in the appendix. (Appendix A. 5)
> > > - Revised the related work section for clarity and completeness. (Related Work)
> > > - Added the “Limitations & Future Work” subsection in the appendix. (Appendix A. 9)

---

### Official Review · Reviewer_Dr8R · 2025-10-27

**Soundness:** 3
**Presentation:** 3
**Contribution:** 3
**Rating:** 6
**Confidence:** 3

**Summary:**

This paper aims to propose a method to unify several important tasks of 3D scene understanding. To this end, the authors propose a probabilistic graphical model, where nodes represent multimodal scene elements such as RGB, optical flow, and camera pose. The combinations of these nodes therefore enables different tasks, including novel view synthesis, 3D object manipulation and depth estimation without task-specific training. Experiments show that the unified model outperforms the respective task-specific baselines. The authors also discover that the model supports composable inference pathways, therefore can help other tasks such as navigation and amodal completion.

**Strengths:**

1. Novelty of the problem. I like the story that the authors want to develop a unified model for different 3D understanding tasks.

2. Novelty of the method. The method to unify the different 3D understanding tasks is also novel and interesting to me.

3. Good evaluation results. Experiment results show the developed unified model generally outperforms the task specific models for novel view synthesis, 3D object manipulation and depth estimation, although it has not been trained for the specific tasks. The results are impressive to me.

**Weaknesses:**

1. Efficiency. It is good to see the authors have developed such a unified model with good performance. It would be better if the authors can also show the efficiency of their model for different tasks compared with the respective task-specific baselines. If the model are going to be applied to embodied AI systems for navigation, for example, the efficiency does matter.

2. Application scenarios. In section 5 of the paper, the authors discuss about the emergent geometric reasoning abilities of the model, and have provided several qualitative examples. However, it increases the paper's contribution and significance if the authors can provide quantitative results on the tasks, such as navigation, and amodal completion, as they mentioned.

**Questions:**

Generally I like the paper. I would recommend the authors to address my concerns in the weakness section during the rebuttal.

---

> ### Author Response · Authors · 2025-11-22
>
> **Efficiency.**
>
> Thank you for raising this important point. We conducted additional measurements comparing our inference time with task-specific baselines under the same hardware conditions. All experiments were performed on NVIDIA A40 GPUs (46 GB) running on dual-socket AMD EPYC 7352 systems (48 physical cores, 96 threads).
>
> For novel view synthesis:
>
> - ViewCrafter: 240 s (25 frames)
> - MotionCtrl: 62 s (14 frames)
> - ZeroNVS: 143 min (NeRF distillation), 2 min rendering (25 frames)
> - SEVA: 10 min 47 s (30 frames)
> - Ours: 253 s (1 frame)
>
> For 3D object manipulation (1 frame):
>
> - LightningDrag: 7.5 s
> - DragAnything: 99 s
> - Diffusion Handles: 135 s
> - Ours: 253 s
>
> For depth estimation from a single image:
>
> - SC-DepthV2: 5 ms
> - IndoorDepth: 79 ms
> - MotionCtrl: 62 s
> - ZeroNVS: 143 min (no quantitative evaluation)
> - SEVA: 10 min 47 s
> - Ours: 140 s
>
> For generative tasks, our runtime is within the typical generative-model range. Importantly, we have not applied dedicated engineering optimizations as this is the first model that performs autoregressive generation of RGB, optical flow, and camera pose within a unified model. We expect standard efficiency techniques used in large autoregressive models can improve the efficiency without conceptual obstacles. For the depth estimation task, we are not aiming to match the efficiency of specialized regression systems. The task remains valuable because it reveals the model’s underlying geometric understanding. If a fast estimator is desired in practice, the geometric knowledge in our unified model can be distilled into a lightweight regression model.
>
>
> **Application Scenarios.**
>
> We appreciate the suggestion to include more quantitative evaluations related to applications such as navigation and amodal completion.
>
> - Navigation.
> We agree that testing navigation or planning abilities would be highly interesting. However, this requires implementing online control and simulator interface, which is not feasible within the rebuttal period. We plan to explore this direction as future work, and we discuss it in the appendix.
>
> - Amodal Completion.
> We added a quantitative amodal completion evaluation to the appendix. Because baseline editing methods can all remove objects, we evaluate how well each model reconstructs the missing depth behind the moved object in 3DEditBench. Specifically, we segment the region where the original object was located and compare the predicted depth in the edited image with the ground-truth depth from the target view, restricted to that region. Depths are estimated using DepthAnything v2 and aligned by median-based scale normalization.
>
> This metric directly reflects a model’s ability to infer the hidden geometry behind an object. Our model shows better depth reconstruction accuracy in this occluded region than the baselines. We have added the full quantitative tables to the appendix.
>
>
> **List of changes in the paper:**
>
> - Added a discussion on navigation as future work in the appendix. (Appendix A. 9)
> - Added a quantitative amodal completion evaluation in the appendix, including depth-based metrics for disoccluded regions and full comparison tables. (Appendix A. 8)

---

### Official Review · Reviewer_AqTG · 2025-11-02

**Soundness:** 3
**Presentation:** 3
**Contribution:** 3
**Rating:** 6
**Confidence:** 4

**Summary:**

The proposed physical world model aims to enable 3D scene understanding enabling improvements in visual reasoning tasks.
The method proposes a probabilistic graphical model operating on scene information such as RGB patches, optical flow, and depth maps, implemented as a 7B autoregressive transformer performing next-token prediction. The model is trained on Big video dataset and 3D vision benchmarks such as Co3D, ScanNet++, and others. The model enables novel-view synthesis, 3D object manipulation, and depth estimation. The paper presents strong results for each of the tasks on standard test benchmarks demonstrating improvements over existing methods. The authors also contribute a new dataset of 100 image pairs for 3D object manipulation which could be useful for future research.

**Strengths:**

The paper is well-motivated and proposes a simple yet effective method enabling strong results for Novel-view synthesis, 3D object manipulation, and depth estimation tasks. The paper expresses the ideas with enough detail for understanding each component and its implementation. The evaluation is comprehensive with existing methods on standard benchmarks for the respective tasks. Composable inference pathways for 3D scene understanding as shown in Figure 6 is a strong result and could be useful for downstream robotics tasks. This can be used for complex reasoning and planning in 3D scenes.

**Weaknesses:**

- The results do not show strong understanding of lighting and appearance understanding based on the results in Figure 3 (the specular highlights on the objects) and 4 (the macbook), as the objects seem to have baked in shading.
- The object manipulations are limited to rigid transformations and do not show compositional understanding of objects such as stacked objects. The demonstrated examples are limited to the new dataset and would require more evidence to support the claim of improved 3D object manipulation capabilities.
- The paper should include a discussion on the limitations of the model to better inform the reader about the scope of the contributions.
Suggestions for improvement of the presentation:
- I recommend the authors to include a short introduction to each section as an overview of the content.
- Introduction can include a concise summary of the contributions of the paper.

Minor:
- L115: an video --> a video

**Questions:**

- Can the model perform transformation of non-rigid objects such as cloth? If so, how can the transformations be performed?

---

> ### Author Response · Authors · 2025-11-22
>
> **“The results do not show strong understanding of lighting and appearance…”**
>
> We agree that the model does not fully capture lighting behavior in some of the cases the reviewer pointed out. At the same time, we have observed many successful examples where the model correctly reasons about shadows and view-dependent appearance of objects. We do not view this as a fundamental limitation of the modeling approach. Rather, we expect lighting fidelity to improve with larger models and more diverse training data. We have added additional generation examples illustrating correct lighting behavior in the revised appendix.
>
> **“The object manipulations are limited to rigid transformations…”**
>
> We acknowledge that the examples in the paper focus on rigid manipulation because the 3DEditBench dataset primarily contains rigid transformations. However, our framework is not limited to rigid object manipulation. Because our tokens are local and the conditioning interface supports both sparse and dense control, the model naturally handles deformable object manipulations. By applying flow conditioning only to the deforming region and leaving the rest unconstrained, the model infers the resulting non-rigid transformation. We have added qualitative examples demonstrating this behavior.
>
> For stacked or multi-object scenes, we have included new examples in which the model manipulates a single object while keeping the others fixed. In these cases, we segment the target object and apply full flow conditioning to that object while leaving the rest of the scene unchanged in the conditioning. This demonstrates the model’s ability to operate on one object among multiple similar ones and maintain overall scene coherence.
>
> **“The paper should include a discussion on the limitations…”**
>
> Thank you for the suggestion. We have added a dedicated “Limitations & Future Work” subsection in the appendix, including illustrative examples of occasional failure cases such as motion-induced blur, object duplication, and sensitivity to segmentation quality during rigid object manipulation. These behaviors occur infrequently and are already reflected in our quantitative metrics, where our model continues to outperform prior work.
>
> **“Suggestions for improvement of the presentation…”**
>
> We incorporated the reviewer’s suggestions by adding short overview paragraphs at the beginning of each major section and including a concise summary of the contributions in the Introduction. We also corrected the minor typo (“an video” → “a video”).
>
>
> **List of changes in the paper:**
>
> - Clarified limitations of lighting and appearance understanding and added qualitative examples in the appendix showing correct handling of shadows and lighting. (Appendix A. 6)
> - Expanded the scope of object manipulation results by adding qualitative examples of non-rigid manipulation using localized flow conditioning. (Appendix A. 7)
> - Added new examples for stacked and multi-object scenes demonstrating that the model can manipulate a single segmented object while keeping the rest of the scene fixed. (Appendix A. 7)
> - Added a dedicated limitations subsection in the appendix. (Appendix A. 9)
> - Improved overall presentation quality by adding overview paragraphs to each major section and a concise contribution summary in the Introduction. (Introduction, Methods, and Results)
> - Corrected minor textual issues, including the change from “an video” to “a video.” (Figure 1 Caption)

---

> > ### Comment · Reviewer_AqTG · 2025-11-26
> >
> > Thanks for addressing the questions and incorporating the suggestions.
> >
> > **Lighting and appearance understanding**: The examples are sufficient and match the decent performance I would have expected from a single image.
> >
> > **Deformation and occlusions**:
> > - The first example for deformable objects is not quite convincing since the cloth is still treated more or less as as a rigid object. Maybe replacing that example with an image of a blanket with a large flow from one area to the other might reveal much more interesting. What was the requirement of the anchor points and what happens if they are not input to the model?
> > - For compositionality, the results are convincing.
> >
> > **Limitations**: I firmly believe that the limitation section should be part of the main manuscript even if the figures referred are attached in the supplementary document.

---

> > > ### Author Response · Authors · 2025-12-04
> > >
> > > **“The first example for deformable objects…”**
> > >
> > > Thank you for pointing this out. We have replaced the previous example with a more representative deformable object case involving a blanket undergoing a non-rigid motion. The updated example reveals a clearer separation between the region that follows the prompted motion and surrounding areas that stay relatively stationary, producing a more plausible non-rigid deformation.
> > >
> > > **“What was the requirement of the anchor points…”**
> > >
> > > A key aspect of our model is that it is probabilistic, and for any given single motion prompt, it will cause different motions, all of which are consistent with the original prompt, as a function of the seed. For some seeds, the motions are smaller and more local, while for others, the motions can be larger. Large motions can happen even with a single motion prompt, and no additional stop patches, as the updated example now shows.
> > >
> > > Another key aspect of our model is that it is densely controllable, and can accept any number of prompt patches, while keeping generations consistent with those multiple prompts. For rigid objects, only one prompt patch is needed to essentially determine the object motion -- in other words, the entropy of the conditional motion distribution conditioned on that prompt is already low. But for deformable objects, a single motion patch prompt will necessarily (physically correctly) determine less about other motions elsewhere on the object -- the entropy of the conditional motion distribution conditioned on that prompt remains high on parts of the object further from the motion prompt location. This entropy being high is the reason why some samples will show more localized motion and others will show more global motion (and is, in fact, a kind of measure of "deformability" in itself).
> > >
> > > In our original example, we give the model two prompt patches (the motion patch and the stop patch) to control the motion more explicitly to have a specific property -- e.g. very local non-rigid deformation. Now in the updated example, because the stop patch has been removed, some samples will have global motion. Hopefully, our examples illustrate that, our system will support different prompts that allow the user to express different goals.
> > >
> > > **“...the limitation section should be part of the main manuscript…”**
> > >
> > > Thank you for the suggestion. We have added a Discussion section to the main manuscript, which now includes the limitations.
> > >
> > > **List of changes in the paper:**
> > > - Updated the deformable object examples in the appendix (Appendix A.7).
> > > - Added a Discussion section in the main manuscript, which now includes the limitations.

---

### Official Review · Reviewer_eRy7 · 2025-11-03

**Soundness:** 3
**Presentation:** 3
**Contribution:** 4
**Rating:** 8
**Confidence:** 2

**Summary:**

The paper presents an autoregressive generative model (3WM) for RGB images and optical flow based on an interesting "pointer-content representation" that allows a single autoregressive model to generate image or optimal flow output from sparse or dense image or optical flow input. Effectively, the paper trains a single model to learn to approximate a diversity of types of conditional distributions on image and optical flow tokens, via randomization of the training set. The paper demonstrates the resulting model on several tasks including novel-view synthesis, 3D object manipulation, and depth extraction.

**Strengths:**

The pointer-content representation that allows for a single model to approximate a number of conditional generation tasks on the joint space of (RGB image, optical flow, RGB image) is significant. The approach of training a model to approximate diverse types of conditional distributions via randomly selecting and re-reorder "pointer-content" pairs is a useful contribution. The qualitative comparisons to baselines shown in the paper are compelling.

**Weaknesses:**

The use of "graphical model" as the formal framework for describing the work seems somewhat ill-suited, since there are no conditional independences in the model (and not surprisingly, the the paper includes no graphical models visualized). See "Questions" for connections to other formal frameworks for generative modeling that seem more aligned with the "pointer-content" representation.

**Questions:**

1. The specific way that training examples were generated from the video dataset seems central to the model. It would help to provide more detail on the type of (latent, observed) pairs that were generated synthetically from the video dataset, and the distribution of these different types of examples at training time?
2. Since the autoregressive model represents in principle all possible conditional distributions, it would be interesting to study how self-coherent it is. It would be interesting to compare (i) the approximate amortized inferences obtained by sampling from the trained autoregressive model, with (ii) the idealized conditional distribution that is defined by sampling from the unconditioned joint distribution and performing rejection sampling. One way to do this is via estimating expected symmetrized KL divergence on synthetic data generated from the model itself as in "AIDE: An algorithm for measuring the accuracy of probabilistic inference algorithms" to compare rejection sampling versus the variational approximation learned by the model (this essentially boils down to sampling from the joint model, then evaluating the conditional probability of the latent tokens given the observed tokens under the autoregressive model). Are there certain classes of queries where it is more self-consistent than others? Can you use the joint model to do a little model-based Monte Carlo (importance sampling) to improve upon proposals sampled from the autoregressive amortized approximation of the conditional distribution?
3.  Interesting connection: The "pointer-content" representation doesn't fit well with standard graphical models, but is more closely related to the address-value representation used in the Gen probabilistic programming language (see e.g. section 4.1 of Gen: A General-Purpose Probabilistic Programming System with Programmable Inference) where random choices are assigned labels ("addresses") and generative models are probability distribution on dictionaries that map addresses to values.

---

> ### Author Response · Authors · 2025-11-22
>
> **“The specific way that training examples were generated from the video dataset…”**
>
> We thank the reviewer for raising this important question about how training sequences are constructed, as this is central to our autoregressive formulation. Each training example is formed as a causal token sequence. All previously generated tokens serve as observed context, and supervision is applied only to the next token, as in standard language-model training. In our implementation, all RGB and flow tokens after the first frame are predicted autoregressively, while RGB₀, camera pose tokens, and pointer tokens are treated as observed context rather than supervision targets. This is a design choice for this work, not a limitation of the framework.
>
> In general, training sequences are drawn from a mixture of RGB-only and RGB+flow subsequences, and we randomly allocate the number of tokens across frames with different modalities and time indices. Within RGB+flow examples, the model naturally encounters several supervision patterns depending on whether flow is the final target or an intermediate prediction used before the next RGB frame. Some sequences terminate with flow (e.g., RGB₀→Flow₀₁ or RGB₀→RGB₁→Flow₁₂), while others place flow mid-sequence and continue autoregressively to RGB (e.g., RGB₀→Flow₀₁→RGB₁ or RGB₀→RGB₁→Flow₁₂→RGB₂). When pose is available, pose tokens are inserted between RGB frames or immediately before flow.
>
>
>
> **“...it would be interesting to study how self-coherent it is.”**
>
> We have experimented with randomizing both the order in which patches are decoded and the random samples we draw from the distribution of possible values when decoding any given patch. We observe that both of these affect the overall generation. In the aggregate, several samples drawn from the model cover a general distribution of several plausible futures, so they diverge on parts of the scene that present uncertainty (such as occluded objects) and maintain consistency in the parts that are certain (such as static fully-visible objects). We plan on experimenting with rollout consistency and guided sampling in future work, and will experiment with AIDE then.

---

> ### Author Response · Authors · 2025-11-22
>
> **“The use of "graphical model" as the formal framework…”**
>
> Indeed, the reviewer is right, in a way we are only explicitly using a part of the PGM formulation.  The main aspect of it that we are using here is the idea of creating a system which, given any state variable, is able to produce the conditional distribution of that variable, conditioned on any combination of other variable(s) in the system.  This is equivalent to an all-to-all influence graph, and we do not use any non-trivial computation about the graph in our work. The PGM idea is really, as the reviewer notes, the combination of two things: a (i) comprehensive variable conditional distribution estimator system and (ii) the use of nontrivial graph structures representing asserted conditional independence relationships.  We only make use of (i) -- a powerful idea in itself. Ideally there would be a name for idea (i) alone, in which case we would be happy to use that label.
> But interestingly, the issue of the effective independence relationships does somewhat appear in our work, implicitly. Here we show that the model can be used to perform several kinds of extractions e.g. of properties like depth. In other work (not part of this submission) we can show that the same kind of predictor can be used to extract other quantities such as optical flow or segmentation. Although it's a little beyond the scope of this paper to show this, all of these extractions have the form of being, essentially, causal inferences in the underlying (all-to-all) PGM corresponding to our model.   In turn, performing these causal inferences ends up identifying conditional dependence (or independence) structures that could be used to put nontrivial graph structure on the outputs.  For example, when we perform segmentation, this is a statement about which flow distributions depend or don't depend on which other ones (the ones within an object segment corresponding to those with strong conditional dependence). This is an interesting topic for further investigation -- and we hope in a future work to be able to formalize these ideas to the point where we could nontrivially use the graph structures.
>
> **“pointer-content representation… is more closely related to the address-value representation used in the Gen probabilistic programming language.”**
>
> That is a really interesting connection! We had not made that particular connection before explicitly ourselves, but it makes sense. It is definitely the case that we think of our model as implementing a kind of learned partial version of the ideas formalized in Gen, and it makes sense that the pointer/contents formalism is a kind of universally useful construct for keeping track of variables in a generative system.
>
> **List of change in the paper:**
> - Added a detailed description of how training examples are constructed, including the distribution of supervision patterns across RGB, flow, and pose tokens. (Appendix A. 10)

---

### Meta-Review · Area_Chair_tStk · 2026-01-06

**Summary:**

The paper received mixed but slightly positive evaluations （8，6，6，4）.

The reviewers are generally supportive of the unified modeling idea, but raise concerns about clarity, evaluation completeness, and the scope of demonstrated capabilities.

**Reviewer Concerns:**

Reviewer eRy7 (8): Largely positive; raises mainly conceptual framing concerns, noting that the “graphical model” terminology may be ill-suited and suggesting clearer connections to alternative generative modeling or probabilistic programming frameworks, as well as more detail on training data construction.

Reviewer AqTG (6): Questions the depth of 3D understanding, particularly lighting and appearance modeling, and notes that object manipulation is limited to rigid transformations; suggests clearer discussion of limitations and stronger evidence for compositional manipulation.

Reviewer Dr8R (6): Supports the unification idea but raises concerns about efficiency and applicability, requesting explicit efficiency comparisons with task-specific baselines and more quantitative evaluation on downstream application scenarios (e.g., navigation, amodal completion).

Reviewer WZBe (4): Focuses on presentation and evaluation gaps, citing unclear articulation of core contributions, lack of an intuitive overview figure, and insufficient ablation studies to fully validate generalization and effectiveness.

**Reviewer Scores:**

The authors provided detailed responses to all reviewers’ comments. Reviewers AqTG and WZBe offered follow-up feedback, and, as noted by both reviewers, the paper would benefit from incorporating these suggestions into the revised version. Overall, the rebuttal is constructive and addresses the main concerns raised during the review process.

---

### Decision · Program_Chairs · 2026-01-26

Accept (Poster)